# PDE Solvers Should Be Local: Fast, Stable Rollouts with Learned Local Stencils

## Abstract

Neural operator models for solving partial differential equations (PDEs) often rely on global mixing mechanisms—such as spectral convolutions or attention—which tend to oversmooth sharp local dynamics and introduce high computational cost. We present FINO, a finite-difference–inspired neural architecture that enforces strict locality while retaining multiscale representational power. FINO replaces fixed finite-difference stencil coefficients with learnable convolutional kernels and evolves states via an explicit, learnable time-stepping scheme. A central Local Operator Block leverage a differential stencil layer, a gating mask, and a linear fuse step to construct adaptive derivative-like local features that propagate forward in time. Embedded in an encoder–decoder with a bottleneck, FINO captures fine-grained local structures while preserving interpretability. We establish (i) a composition error bound linking one-step approximation error to stable long-horizon rollouts under a Lipschitz condition, and (ii) a universal approximation theorem for discrete time-stepped PDE dynamics. (iii) Across six benchmarks and a climate modelling task, FINO achieves up to 44% lower error and up to around 2× speedups over state-of-the-art operator-learning baselines, demonstrating that strict locality with learnable time-stepping yields an accurate and scalable foundation for neural PDE solvers.

## 1 Introduction

Partial Differential Equations (PDEs) are fundamental to applied mathematics and engineering, governing phenomena in fluid dynamics (John & Anderson, 1995), heat conduction, electromagnetism, structural mechanics, and biology (Edelstein-Keshet, 2005). They describe the evolution of state variables in space and time, enabling prediction, control, and optimisation of complex systems. Analytic solutions for non-linear PDEs are rare; therefore, numerical methods such as finite difference, finite element (Quarteroni et al., 2006) and finite volume have been developed.

Classical solvers have been studied for more than a century, but remain limited by the trade-off between accuracy and efficiency (LeVeque, 2007). Finer discretisations improve accuracy but increase computational cost. Deep learning offers a new route, with two dominant directions. (i) **Physics-informed methods** such as PINNs (Raissi et al., 2019) embed PDE equations directly into the loss, leveraging physical structure to reduce labeled data requirements. However, they often face optimisation difficulties, especially in high-dimensional settings, and perform poorly on solutions with discontinuities, sharp gradients, or symmetries. (ii) **Operator-learning** methods approximate infinite-dimensional operators from data, learning families of PDEs. Training is expensive, but inference is fast. DeepONet (Lu et al., 2019) and FNO (Li et al., 2020a) are notable examples.

Global operator methods such as FNO capture long-range structure but oversmooth local dynamics. **Locality is critical**: for hyperbolic PDEs, finite-speed propagation implies that solutions depend only on data within local characteristic regions (LeVeque, 1992). Local kernels can represent these dynamics efficiently, whereas expressing them globally increases parameter counts. Although spectral methods like FNO and SFNO can, in theory, approximate local behavior, the uncertainty principle forces high parameter counts when representing fine-scale features.

While local operators can be represented using global bases (e.g., Fourier or attention), this comes at a high cost. Capturing localised features requires resolving high frequencies across the entire domain, and the uncertainty principle implies that such global approximations demand dense spectral

representations. Methods like FNO and SFNO must reconstruct the full signal in frequency space, even when the operator is inherently local—leading to high parameter counts and loss of a locality bias that many PDEs naturally exhibit.

Recent advances in operator learning follow four main directions, each with trade-offs. (i) Transformer-based operators capture long-range dependencies via self-attention, but incur quadratic complexity and offer limited operator-theoretic guarantees. (ii) Graph Neural Operators (GNOs) (Li et al., 2020b) learn local kernels, but pairwise kernel evaluation is expensive compared to optimised convolutions, and lacks natural equivariance. (iii) Hybrid U-former designs (Wen et al., 2022) inject local priors into global architectures by combining CNNs and FNOs, but suffer from high training overhead and fixed-resolution bottlenecks that suppress high-frequency content. (iv) Localised-kernel operators (Liu-Schiaffini et al., 2024) extend FNOs with differential and discretisation-agnostic branches (e.g., DISCO), improving local expressivity at the cost of increased computation, resolution-sensitive runtime, and stacked discretisation error.

These limitations highlight the need for architectures with an intrinsic local bias that remain efficient and accurate. The central challenge is to design models that capture fine local features via strictly local receptive fields—without sacrificing speed or scalability. Such methods are essential for: (1) **Fast computation.** Many PDE applications—such as weather forecasting, robotics, and digital-twin systems—require frequent retraining or real-time inference. This makes fast, efficient computation essential for practical deployment. (2) **Capturing local properties** Many PDEs inherently rely on local interactions, and failing to capture these can significantly degrade accuracy. (3) **Generality** Global operator methods often perform poorly on time-independent PDEs, limiting their applicability. (4) **Accuracy** High precision approximation of fine-scale structures is necessary to ensure predictive fidelity across diverse PDE families.

**Contributions.** We propose **Finite-difference inspired Neural Operator (FINO)**, a neural architecture explicitly inspired by classical finite-difference (FD) schemes. Like traditional FD methods, FINO discretises the domain into local stencils, but replaces fixed coefficients with learnable convolutional kernels. At its core, a *Local Operator Block* learns differentiable operators on each stencil, enforcing a strictly local receptive field and providing a direct correspondence to finite-difference operators. These learned derivatives are advanced in time using an explicit, learnable time-stepping update, ensuring interpretability and stability. The resulting FINO block is embedded within an encoder–decoder structure with down-sampling, skip connections, and up-sampling, enabling the model to capture multiscale features and reconstruct complete solution trajectories. Unlike recent *local-operator* networks, FINO maintains strict locality through stencil-style derivatives and explicit time-stepping. This avoids global spectral transforms and preserves both speed and interpretability. Across benchmarks, this design FINO consistently yields faster training, lower inference time, and higher accuracy than state-of-the-art operator-learning methods. Notable, we emphasise:

- We introduce FINO, which replaces fixed stencil coefficients with learnable convolutional kernels and couples them with an explicit, learnable time-stepping scheme, enforcing strictly local receptive fields while preserving multiscale capacity.

- We prove that FINO is a universal approximator for discrete-time PDE dynamics. We also derive a novel error-propagation bound, showing how local approximation error controls long-horizon rollout stability under mild Lipschitz conditions.

- We validate FINO on six PDEBench benchmarks (1D advection, diffusion–reaction, compressible Navier–Stokes; 2D Darcy flow, diffusion–reaction, shallow water) and a climate modelling task, where it achieves higher accuracy and significantly faster training than state-of-the-art baselines.

## 2 RELATED WORK

- **Deep Learning for PDEs.** Early deep learning approaches for PDEs, such as PINNs (Raissi et al., 2019), embed governing equations into the loss function but struggle with generalisation across resolutions and domains. Neural operators address this limitation by learning mappings between function spaces, yielding discretisation-invariant surrogates. DeepONet (Lu et al., 2019) and FNO (Li et al., 2020a) laid the groundwork for neural operators, providing universal approximation guarantees and efficient spectral modelling of global dependencies. Building on these foundational works, several FNO variants (Tran et al., 2021; Xiao et al., 2024; Park et al., 2025) have been proposed

to further enhance performance, with extensions to adaptive and geometry-aware settings. In parallel, graph-based methods such as the Graph Neural Operator (Li et al., 2020b; Brandstetter et al., 2022) generalise operator learning to non-regular grids, while other directions exploit wavelets, spectral bases, or integral kernels for multiscale and discretisation-robust approximations (Tripura & Chakraborty, 2022; Fanaskov & Oseledets, 2023).

More recently, attention-based architectures have become increasingly popular in operator learning. Transformer-style models such as OFormer (Li et al., 2022), GNOT (Hao et al., 2023), and Transolver (Wu et al., 2024) adapt self-attention mechanisms to capture long-range spatial interactions in PDEs. Within this landscape, HAMLET (Bryutkin et al., 2024) extends neural operators to irregular geometries through graph transformers, radius-based neighborhoods, and cross-attention for arbitrary queries, achieving strong results on PDEBench and Airfoil benchmarks.The Mamba Neural Operator (Cheng et al., 2024) offers a state-space alternative to transformers, improving computational efficiency while retaining accuracy. Therefore, these developments allow the trajectory from grid-restricted spectral methods toward geometry-flexible, transformer-style operator learners with increasing scalability and robustness.

● **Local Neural Operators.** Early works such as PDE-Net (Long et al., 2018) and PDE-Net 2.0 (Long et al., 2019) introduced learnable kernels to directly identify underlying PDEs from data. However, these models were not intended for solving PDEs directly—instead, they focused on identifying the underlying PDE equations from observed data. More recent efforts emphasize incorporating locality into operator architectures to better resolve fine-scale dynamics and enforce physical inductive biases. While global methods such as FNO demonstrate strong performance in many settings, they often suffer from over-smoothing and fail to resolve local details. To address these limitations, local neural operators have been proposed, introducing locally supported kernels that align with the inherent locality of many PDEs, such as hyperbolic systems with finite propagation speeds. For example, Ye et al. (2024; 2022) augmented FNOs with convolutional layers to embed locality, while Wen et al. (2023) combined U-Nets with FNOs for improved multi-scale representations. Similarly, convolutional neural operators (CNOs) (Raonic et al., 2023) exploit convolutional inductive biases but remain constrained by equidistant grid discretisations and the need for downsampling, which risks discarding high-frequency information. To overcome this issue, Liu-Schiaffini et al. (2024) introduced localised integral and differential kernels, further enhancing performance. However, most of these approaches rely on hybrid architectures that combine local and global operators, which increases the training time.

**Existing works & comparison to ours.** Our work (FINO) is strictly local by construction. It eliminates global mixing entirely, using only finite-support convolutional stencils and an explicit, learnable time-stepping scheme. This design enforces compact receptive fields, aligns naturally with the finite-speed propagation of many PDEs (e.g., hyperbolic systems), and avoids the spectral inefficiencies and high parameter counts induced by the uncertainty principle.

## 3 FINITE-DIFFERENCE INSPIRED NEURAL OPERATOR (FINO)

**Problem Statement.** We consider time-dependent partial differential equations (PDEs) whose solutions are vector-valued functions $\mathbf{v} \colon \mathcal{T} \times \mathcal{S} \times \Theta \to \mathbb{R}^d$, where $\mathcal{T} \subset \mathbb{R}$ denotes time, $\mathcal{S} \subset \mathbb{R}^n$ is a spatial domain, and $\Theta$ represents a space of PDE-specific parameters (e.g., coefficients, boundary conditions). For example, in a heat conduction problem, $\mathbf{v}(t, \mathbf{s}, \theta)$ may denote temperature at time $t$, location $\mathbf{s} \in \mathcal{S}$, under conductivity profile $\theta \in \Theta$. We define the forward operator $\mathcal{F}_\theta$ as a parameterised time evolution map that advances the solution by one time step: $\mathcal{F}_\theta \colon \mathbf{v}(t-\ell{:}t-1, \cdot) \mapsto \mathbf{v}(t, \cdot)$, where $\ell$ is the number of previous steps required to estimate temporal derivatives (e.g., for explicit schemes). The discretised version $\mathring{\mathcal{F}}_\theta$ is obtained via a high-resolution numerical solver, such as finite difference or finite volume methods. Our goal is to learn a data-driven emulator $\widehat{\mathcal{F}}_{\theta,\phi} \approx \mathring{\mathcal{F}}_\theta$, parameterised by $\phi$, that generalises across different PDE parameters $\theta$. Given a dataset of $K$ simulated solution trajectories $\mathcal{D} = \left\{ \mathbf{v}_{\theta_k}^{(k)}(0{:}t_{\max}, \cdot) \right\}_{k=1}^{K}$, we estimate $\phi$ by minimising a supervised rollout loss that penalises discrepancies between the predicted and ground-truth states:

$$\widehat{\phi} = \arg\min_\phi \sum_{k=1}^{K} \sum_{t=1}^{t_{\max}} \mathcal{L}\left( \widehat{\mathcal{F}}_{\theta_k,\phi} \{ \mathbf{v}_{\theta_k}^{(k)}(t-\ell{:}t-1, \cdot) \}, \mathbf{v}_{\theta_k}^{(k)}(t, \cdot) \right),$$

where $\mathcal{L}(\cdot,\cdot)$ is a suitable loss function (e.g., mean squared error) comparing predicted and true solutions over the spatial domain. Here, $\{t-\ell,\ldots,t-1\}$ denotes the past $\ell$ time steps used as input to predict the next state at time $t$.

## 3.1 THE ANATOMY OF FINO

FINO is motivated by classical finite-difference (FD) schemes and consists of four main components: (1) a Local Operator Block that mimics local differential operators, (2) an explicit time-stepping update, (3) composable neural blocks forming the operator core, and (4) a U-Net–style encoder–decoder for multiscale modelling. Figure 3.1 provides a visual overview. The architectural flow is detailed next.

**Local Operator Block (LOB).** The Local Operator Block (LOB) is designed to approximate spatial differential operators in a strictly local and learnable manner. It draws direct inspiration from classical finite-difference (FD) methods, where spatial derivatives are estimated using fixed-weight stencils. In contrast, LOB replaces these fixed coefficients with learnable convolutional filters, enabling adaptive, data-driven computation of local derivatives.

The LOB consists of three main components: a learned stencil operation, a gating mechanism, and a fusion step. The first component performs a learnable convolution that mimics finite-difference stencils. Given a 4D input tensor $\mathbf{u} \in \mathbb{R}^{b \times i \times H \times W}$, where $b$ is the batch size, $i$ is the number of input channels, and $H \times W$ is the spatial domain, we define the stencil operator $S(\mathbf{u})$ as:

$$S(\mathbf{u}) = \sum_{c=1}^{i} \sum_{p=-r}^{r} \sum_{q=-r}^{r} w_{\alpha,c,p,q}\, \mathbf{u}_{b,c,(h+p),(w+q)} + b_s \tag{1}$$

This equation defines a convolution with a $(2r+1) \times (2r+1)$ kernel centered at each spatial location $(h, w)$, where $r$ is the stencil radius. The weights $w_{\alpha,c,p,q}$ are learnable parameters, and $b_s$ is a bias term. Compared to traditional finite-difference methods (which use fixed stencils like $[1, -2, 1]$), this formulation enables the network to learn its own approximation of local derivatives from data.

Not all local derivatives are equally relevant for solving a given PDE. To adaptively select the most informative derivative features, we introduce a gating mask, computed as:

$$G(\mathbf{u}) = \sigma\big(W_g * S(\mathbf{u})\big) \odot S(\mathbf{u}) \tag{2}$$

Here, $W_g$ is a learnable convolutional filter, $*$ denotes convolution, and $\sigma(\cdot)$ is the sigmoid function. The gating mechanism produces a per-location, per-channel importance mask (values in $[0, 1]$) that modulates the stencil output via element-wise multiplication ($\odot$). This allows the network to suppress irrelevant or noisy derivative responses and focus on spatial patterns most useful for the current PDE.

The final output of the LOB is computed by fusing the gated features using another learnable convolution:

$$\partial_t U_t = W_c * G(\mathbf{u}) \tag{3}$$

This step linearly combines the gated stencil responses to form the estimated spatial derivative $\partial_t U_t$, which acts as the core update direction for the PDE. The learnable fusion allows the model to leverage directional or cross-derivative terms and synthesise them into a cohesive output.

**Time Integration Scheme.** To advance the solution in time, we adopt an explicit forward Euler scheme:

$$U_{t+\Delta t} = U_t + \Delta t\, \partial_t U_t, \quad \Delta t = \theta \in \mathbb{R}_{>0} \tag{4}$$

where $\partial_t U_t$ is the learned update produced by the previous local operator block, and $\Delta t$ is a learnable scalar parameter shared across the domain. The time-step parameter is initialised to a small positive value and updated during training. This formulation preserves the causal structure of temporal evolution and allows the network to learn time dynamics explicitly. Unlike autoregressive models that directly regress the next frame, this update mimics classical numerical solvers by estimating a derivative and applying it via time integration, offering interpretability and numerical grounding.

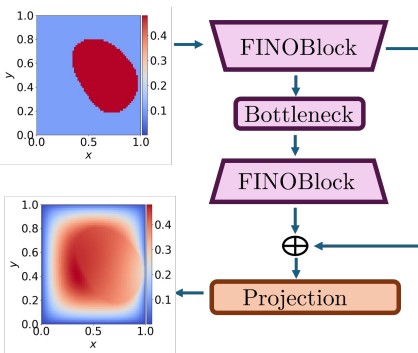

Figure 1: **FINO Framework.** Encoders down-sample and decoders up-sample, with skip connections preserving original feature information.

**FINO Block.** The FINO Block encapsulates the explicit time update from the previous step and enhances it through nonlinear local transformations. Specifically, given the updated state from time evolution, we apply a convolutional layer followed by a ReLU activation:

$$\mathcal{B}(U_t) = \text{ReLU}\Big(W_p * \big[ U_t + \Delta t\, \partial_t U_t \big]\Big). \qquad (5)$$

where $W_p$ is a learnable convolutional kernel. While the optimal depth is PDE-dependent, we find that using up to three stacked FINO blocks provides sufficient capacity for the range of PDEs considered. To increase modelling capacity, we stack multiple such blocks:

$$\text{FINOBlockStack}(u) = \mathcal{B} \circ \cdots \circ \mathcal{B} \circ \mathcal{B}\,(u) \qquad (6)$$

**FINO Architecture.** To capture both fine-grained local patterns and broader contextual information, FINO adopts a U-Net–style encoder–decoder architecture.

The encoder progressively downsamples the feature maps using:

$$\mathbf{x}_{i+1} = \mathcal{P}_\downarrow(\mathcal{D}_i(\mathbf{x}_i)) \qquad (7)$$

where $\mathcal{D}_i$ applies FINO Blocks and $\mathcal{P}_\downarrow$ is a downsampling operation such as average pooling. At the bottleneck, we compute:

$$\mathbf{z} = \mathcal{K}(\mathbf{x}_N), \quad \text{with} \quad \mathcal{K} := \text{FINOBlockStack} \qquad (8)$$

The decoder then upsamples and fuses features from the encoder via skip connections:

$$\mathbf{x}_{i-1} = \mathcal{U}(\mathbf{z}) + \mathbf{x}_i, \quad \text{for } i = N, \dots, 1 \qquad (9)$$

Finally, the output is projected back to the desired dimensionality using a $1 \times 1$ convolution:

$$\mathbf{y} \in \mathbb{R}^{B \times H \times W \times \text{out\_channels}}$$

## 3.2 THEORETICAL FOUNDATIONS OF FINO

In this section, we first demonstrate that FINO is a universal approximator and discuss its significance. To establish this result, we introduce the following lemma and proposition. The complete proof and statement of Proposition 1 and Theorem 2 can be found in the Appendix B .

**Proposition 1** (Informal, Local-to-Global Error Bound). *If a surrogate map $\Psi_\theta$ uniformly approximates the true PDE one-step map $\Phi_{\Delta t}$ within tolerance $\varepsilon'$, i.e.*

$$\|\Psi_\theta(u) - \Phi_{\Delta t}(u)\| \le \varepsilon' \qquad \forall u, \qquad (10)$$

*and if $\Phi_{\Delta t}$ is Lipschitz with constant $C$, then after $K$ time steps we have*

$$\big\|(\Psi_\theta)^K(u_0) - (\Phi_{\Delta t})^K(u_0)\big\| \le \begin{cases} \dfrac{C^K - 1}{C - 1}\, \varepsilon', & C \ne 1, \\[2mm] K\,\varepsilon', & C = 1. \end{cases}$$

**Theorem 2** (Universal Approximation of FINO for Discrete Time–Stepped PDE Dynamics ). *For a final time $T = K\Delta t$ with integer $K \ge 1$, define the exact solution after $K$ steps as*

$$u(T) = \big(\Phi_{\Delta t}\big)^K(u_0),$$

*where $(\Phi_{\Delta t})^K$ denotes the $K$–fold composition.*

*Then for every compact set $\mathcal{U} \subset X$ and every tolerance $\varepsilon > 0$, there exists a depth–$K$ FD–NET*

$$\Psi_\theta^{(K)} := \underbrace{\Psi_\theta \circ \cdots \circ \Psi_\theta}_{K \text{ identical blocks}},$$

*such that*

$$\sup_{u_0 \in \mathcal{U}} \big\| \Psi_\theta^{(K)}(u_0) - \big(\Phi_{\Delta t}\big)^K(u_0)\big\| \le \varepsilon.$$

Proposition 1 is used to bridge the gap between *local error* and *global error* in the PDE approximation. While the classical universal approximation theorem guarantees that a neural network can approximate a single continuous map, such as the one-step evolution operator $\Phi_{\Delta t}$, to within some tolerance $\varepsilon'$, solving a PDE requires iterating this operator $K$ times to reach the final time $T = K\Delta t$. Without an additional control mechanism, these small local errors could accumulate and potentially blow up over multiple iterations. The proposition provides precisely this control by giving an error propagation formula: it shows that the global error after $K$ steps is bounded by a geometric factor depending on the Lipschitz constant $C$. This ensures stability of the approximation, meaning that as long as $C$ is moderate—or even less than one in the case of dissipative PDEs—the total accumulated error remains controlled rather than diverging with the number of steps.

Theorem 2 –the composition error bound –elevates a one–step approximation guarantee into a long–horizon stability claim for FINO's autoregressive rollouts. Concretely, if the exact one–step propagator $\Phi_{\Delta t}$ is Lipschitz with constant $C$ and the learned surrogate $\Psi_\theta$ matches it within $\varepsilon'$, then after $K$ steps the total error is bounded by $\frac{C^K-1}{C-1}\varepsilon'$ (or $K\varepsilon'$ when $C = 1$). This prevents small per–step discrepancies from snowballing, providing the missing theoretical link between minimizing the stepwise training loss and achieving reliable multi–step predictions. In practice, it justifies the paper's explicit time–stepping with strictly local learned stencils: tightening the per–step fit provably tightens end–to–end trajectory error, thereby grounding the stable long–horizon rollouts in a precise stability mechanism.

**Training Scheme.** We adopt an autoregressive training scheme. At time $t$, we assemble features by concatenating the last $K$ ground-truth (or rolled) frames with the spatial coordinates $G(x,y)$, yielding

$$X_t(x,y) = \text{concat}\big(u_{t-K+1:t}(x,y), G(x,y)\big) \in \mathbb{R}^{KV+2}. \tag{11}$$

A convolutional predictor $f_\theta$ maps these features to the next-frame estimate, $\hat{u}_{t+1} = f_\theta(X_t)$. Let $x^{(0)} = u_{0:K}$ denote the initial rolling buffer of $K$ frames. For $t = K, \ldots, K + H - 1$ (horizon $H$), we iteratively predict and update the buffer via

$$\hat{u}_t = f_\theta\big(\text{concat}(x^{(t-K)}, G)\big),$$
$$x^{(t-K+1)} = \text{shift}\big(x^{(t-K)}\big) \parallel \hat{u}_t, \tag{13}$$

where $\text{shift}(\cdot)$ discards the oldest frame and appends the newest prediction (and $\parallel$ denotes concatenation along the time dimension).

**Loss Function.** We minimise the stepwise mean-squared error accumulated across the full rollout and backpropagate through the entire unroll:

$$\mathcal{L}_{\text{step}}(\theta) = \sum_{t=K-1}^{K+H-1} \frac{1}{B} \sum_{b=1}^{B} \left\| \hat{u}_{t+1}^{(b)} - u_{t+1}^{(b)} \right\|_2^2, \tag{14}$$

where $B$ is the batch size and $\| \cdot \|_2$ denotes the Euclidean norm over all spatial and channel entries. For monitoring and model selection, we additionally report the full-trajectory MSE over the evaluation window,

$$\mathcal{L}_{\text{full}}(\theta) = \frac{1}{B} \sum_{b=1}^{B} \left\| \hat{U}_{0:K+H}^{(b)} - U_{0:K+H}^{(b)} \right\|_2^2, \tag{15}$$

where $U_{0:K+H}$ and $\hat{U}_{0:K+H}$ stack the ground-truth and predicted sequences, respectively. As per our setup, only $\mathcal{L}_{\text{step}}$ contributes to the loss value, while $\mathcal{L}_{\text{full}}$ is used for validation.

## 4 EXPERIMENTS

We provide the details of the dataset, implementation, experiment results, and ablation studies.

### 4.1 DATASET AND IMPLEMENTATION DETAILS

**Datasets.** We conducted experiments on a broad range of PDEs drawn from the PDEBench (Takamoto et al., 2022) and Climte Modelling (Kissas et al., 2022). Specifically, our evaluation covered 1D problems including advection, diffusion–reaction, and compressible Navier–Stokes equations, as well as 2D problems such as Darcy flow, diffusion–reaction, and shallow water equations.

Table 1: Performance Evaluation of FINO Against Global, Local, and Transformer-Based Operators on PDEBench Benchmarks and Climte Modelling. The RMSE reported the results. The best-performing results are highlighted in green , while the second-best results are indicated in bold.

| Dataset | U–Net | FNO | UFNO | FFNO | Transolver | LocalFNO | FNIO (ours) | Improvement |
|---|---|---|---|---|---|---|---|---|
| Advection (1D) | 0.05257 | **0.00530** | 0.00968 | 0.00683 | 0.00937 | - | 0.00296 | 44.15 % |
| CNS (1D) | 12.56934 | 0.34053 | 0.44692 | **0.29493** | 0.72094 | - | 0.1946 | 34.02% |
| Diffusion-Reaction (1D) | 0.03812 | 0.02884 | 0.00891 | 0.01409 | **0.00686** | – | 0.00575 | 16.18% |
| Darcy Flow (2D) | 0.01117 | 0.02575 | 0.06929 | 0.02482 | 0.01987 | 0.02051 | **0.01158** | -3.67% |
| Diffusion Reaction (2D) | 0.05979 | 0.01055 | 0.01443 | 0.01314 | 0.01485 | **0.00346** | 0.00214 | 38.15 % |
| Shallow Water (2D) | 0.11974 | 0.01448 | 0.01500 | 0.01191 | 0.00586 | **0.00438** | 0.00259 | 40.87 % |
| Climate Modelling (2D) | 1.0126 | 0.01536 | **0.00810** | - | 0.18575 | 0.01508 | 0.00715 | 11.73 % |

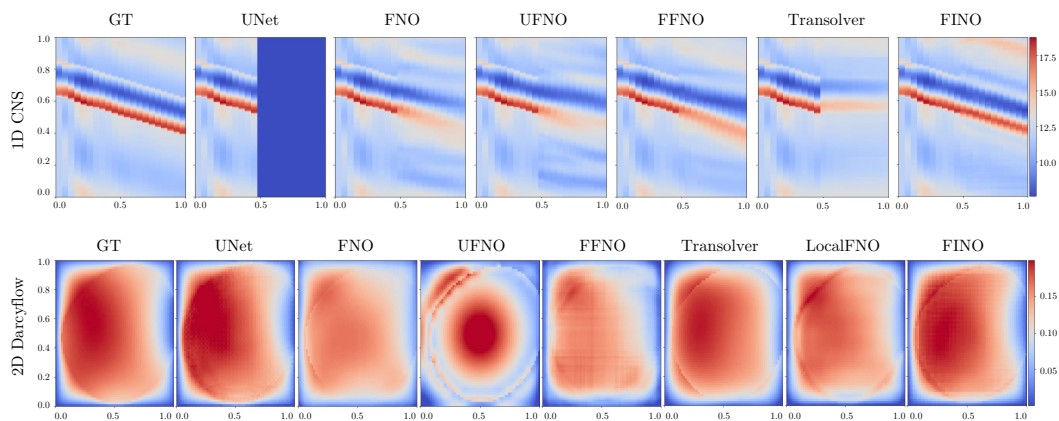

Figure 2: Visual comparison of baseline methods and FINO on 1D CNS and 2D Darcyflow.

In addition, we evaluated our method on a climate modelling dataset. Further details about the datasets are provided in the Appendix C.1.

**Baselines and Training details.** We compared our method against three representative categories of approaches: (i) **Global operator methods:** FNO (Li et al., 2020a), FFNO (Tran et al., 2021); (ii) **Local operator + Global methods:** U-Net (Ronneberger et al., 2015), UFNO (Wen et al., 2022), Local FNO (Liu-Schiaffini et al., 2024); (iii) **Transformer-based methods:** Transolver (Wu et al., 2024). All models were trained for 400 epochs using a single NVIDIA A100 40GB GPU and the climate modelling was trained for 500 epochs. We followed the default training protocols from PDEBench unless otherwise specified. We report the RMSE in Table 1.

## 4.2 EXPERIMENT RESULTS

**Numerical Results** Table 1 shows numerical result in all datasets. For 1D PDEs, FINO consistently achieves the lowest error across all three datasets, reflecting its design as a local operator method. These problems are characterized by strong local structures, where pointwise updates are driven by local derivatives, and FINO benefits directly from its purely local design. In the Advection equation, which is dominated by sharp, locally transported features, FINO achieves the best RMSE (0.00296), improving on the strongest baseline (FNO, 0.00530) by 44.15%. This substantial margin highlights the strong alignment between FINO's locality and the underlying transport dynamics. In the more complex CNS dataset, FINO again obtains the lowest RMSE (0.1946), outperforming the best competing method (FFNO, 0.29493) by 34.02%, indicating that local update rules remain advantageous even for challenging 1D fluid dynamics. Similarly, in the Diffusion–Reaction system, FINO achieves the best performance (0.00575), improving on Transolver (0.00686) by 16.18%, demonstrating robustness in coupled local processes such as diffusion and reaction. Overall, across all 1D tasks, FINO delivers consistent gains, with the most pronounced improvements observed in problems governed by strong local transport mechanisms. Its finite-difference-inspired locality translates directly into more accurate step-by-step updates and significantly lower RMSE.

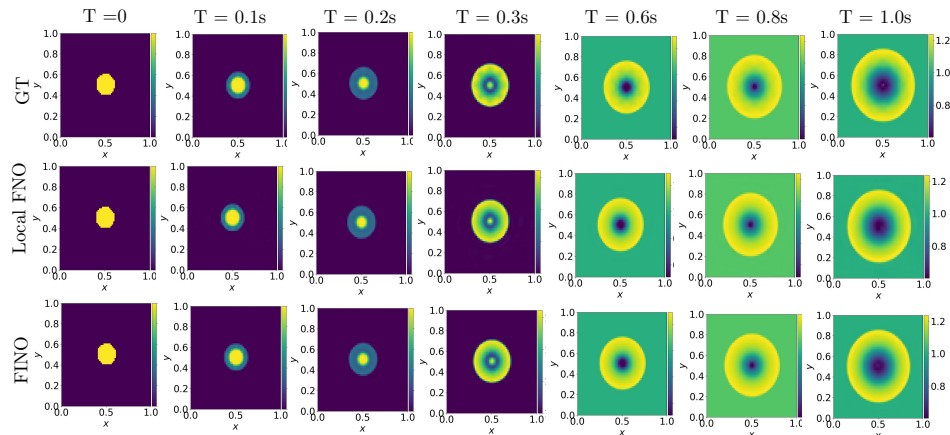

Figure 3: Spatiotemporal comparison of LocalFNO and FINO predictions on the 2D Shallow Water equation benchmark. Each row shows model rollouts at successive time steps.

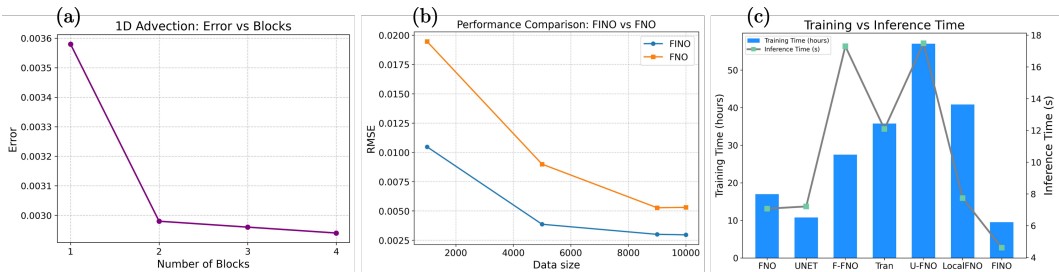

Figure 4: Comprehensive evaluation of FINO compared to baseline operator networks. (a): Error vs. number of composition blocks for the 1D advection task. (b): Data-scaling performance comparison of FINO and FNO. (c): Training vs. inference time across architectures.

For the 2D tasks, we distinguish between time-independent and time-dependent datasets. On the time-independent Darcy Flow benchmark, all global spectral/transform models underperform, while the purely local U-Net achieves the best RMSE (0.01117). FINO is very close to U-Net with 0.01158 (–3.67% relative to U-Net), indicating that local operators are more effective than global ones in steady-state elliptic problems where global mixers tend to oversmooth fine-scale heterogeneity. In contrast, across all time-dependent 2D datasets, U-Net becomes the worst-performing baseline, while FINO consistently achieves the best results: Diffusion–Reaction (0.00214, 38.15% better than LocalFNO) and Shallow Water (0.00259, 40.87% better than LocalFNO). In the climate modelling task, FINO (0.00715) outperforms UFNO (0.00810) by 11.73% because climate evolution is dominated by local advection–diffusion updates. Global models tend to oversmooth sharp gradients, while FINO's local time stepping preserves fine-scale features and controls error growth. Unlike U-Net, which lacks an explicit temporal update, FINO structured locality ensures more stable and accurate rollouts in time-dependent dynamics. These results from 2D time-dependent PDE highlight that time-dependent dynamics benefit from operators that implement accurate local temporal updates grounded in PDE structure. In Appendix C.2, we report additional evaluation metrics. From the data perspective, we include normalized RMSE (nRMSE) and maximum error. From the physics perspective, we present the RMSE of conserved value, RMSE of Fourier-space in low, medium, high-frequency regimes.

**Visualisation Results.** Figure 2 shows the visualisation results on 1D CNS and 2D Da. FINO preserves sharp structures in CNS and produces smooth, consistent Darcy fields, closely matching the ground truth. Unlike global methods, which succeed in CNS but fail in Darcy flow, FINO performs well across both time-dependent and time-independent PDEs. Figure 3 Spatiotemporal rollout comparison on a 2D time dependent PDE. Each column shows the solution field at different time steps

(T = 0s to 1.0s), while rows correspond to the ground truth (GT), Local FNO, and our proposed FINO. FINO produces stable long-horizon predictions that remain visually and quantitatively consistent with the ground truth, demonstrating its improved accuracy and robustness in modeling PDE dynamics. Figure 5 compares the ground truth and predicted global surface pressure fields. FINO produces reconstructions that closely align with the reference, accurately capturing large-scale spatial variations and preserving fine regional structures without introducing spurious artifacts. The smoothness and consistency of the predicted fields highlight the model's robustness in handling climate-scale PDE data, demonstrating its ability to generalize to complex, real-world geophysical patterns. More visualization can be found in Appendix C.2.

## 4.3 ABLATION STUDIES

**Number of FINO Blocks.** Figure 4 (a) shows how the accuracy increases as the composition of FINO blocks in the FINO increases for the 1D advection PDE. A single block yields the lowest, but as the model depth grows to four blocks, the error steadily drops to 0.00294. This indicates that additional depth enhances the model's capacity to capture fine-grained local dynamics, while gains begin to saturate after two blocks.

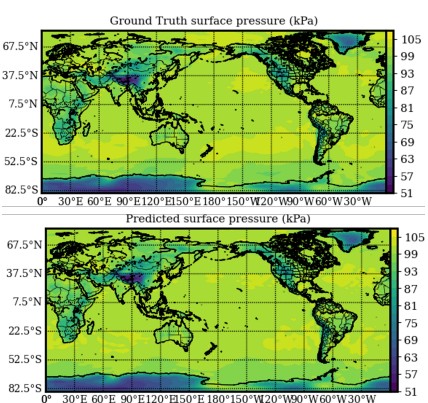

Figure 5: Ground truth (top) and predicted (bottom) global surface pressure. FINO accurately reconstructs large-scale patterns and regional variations, producing smooth, consistent fields that closely match the reference.

**Data Size.** Figure 4 (b) compares FINO and FNO among different dataset size (1k, 5k, 9k, 10k). As the dataset grows, both models reduce their RMSE. However, FINO consistently outperforms FNO across all sizes, and its advantage is particularly pronounced in low-data regimes (1k–5k samples), highlighting FINO's stronger data efficiency and generalization under data scarcity.

**Training and Inference Time.** Figure 4 (c) presents a comparison of training and inference times across all evaluated architectures. Transformer-based methods and U-FNO exhibit significantly higher computational cost, requiring prolonged training durations and slower inference speeds—often making them impractical for deployment or iterative scientific workflows. In contrast, **FINO achieves the fastest inference time**, clocking in at just 4–5 seconds per evaluation, and maintains competitive training efficiency relative to FNO and LocalFNO. This remarkable speed advantage stems from FINO's strictly local convolutional design and lightweight architecture, which avoids the overhead of global attention or spectral transforms. These results highlight FINO's suitability for real-world use cases where rapid model execution and retraining are critical—such as in-the-loop simulations, uncertainty quantification, or interactive PDE exploration. FINO offers an attractive balance between accuracy and computational efficiency

## 5 CONCLUSION

We introduced FINO, a neural operator framework inspired by classical finite-difference schemes, which leverages local learned stencils and explicit time integration to model PDE dynamics efficiently and interpretably. Our work reinforces a central premise: *locality matters* in neural PDE solvers. By constraining the architecture to use compact, learnable spatial stencils and an explicit forward Euler scheme, FINO retains interpretability while achieving strong empirical and theoretical performance. Our theoretical analysis establishes connections between local approximation error and global rollout stability, offering provable guarantees for long-horizon predictions. Empirically, FINO consistently outperforms competitive baselines—including global operators like FNO and Transformer-based models—across several datasets. Notably, FINO achieves up to 44% lower RMSE and 2× faster inference. These results suggest that hybridising classical numerical insights with modern learning yields principled, efficient, and generalisable PDE surrogates. FINO offers a scalable blueprint for designing neural operators grounded in locality, stability, and interpretability.

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

# APPENDIX

In this appendix, we provide additional details regarding our methodology and a more comprehensive description of the dataset used in our experiments.

## A  SUPPLEMENTARY INFORMATION

### A.1  PRELIMINARIES: FINITE DIFFERENCE METHOD

In this section, we provide a brief introduction to the finite difference method (FDM). Partial Differential Equations (PDEs) are inherently complex, requiring various advanced numerical methods. One classical approach is the finite difference method, which approximates partial derivatives by converting them into arithmetic operations (addition, subtraction, multiplication, and division) applied to discrete function values sampled on a computational grid. In numerical analysis for PDEs, a stencil is a structured set of points around a specific node used to approximate derivatives and other key quantities. Stencils underpin many numerical PDE methods, such as the five-point stencil for second-order spatial derivatives and the Crank–Nicolson stencil for time-dependent problems.

Finite difference methods approximate partial differential equations (PDEs) by replacing derivatives with linear combinations of function values on a discrete grid. For instance, a central difference approximation of the first derivative in the $x$-direction at $(i,j)$ is given by:

$$\left.\frac{\partial u}{\partial x}\right|_{i,j} \approx \frac{u_{i+1,j} - u_{i-1,j}}{2\,\Delta x}, \tag{S.1}$$

and the second derivative by:

$$\left.\frac{\partial^2 u}{\partial x^2}\right|_{i,j} \approx \frac{u_{i+1,j} - 2u_{i,j} + u_{i-1,j}}{\Delta x^2}. \tag{S.2}$$

Higher accuracy requires larger stencils. For example, a fourth-order central difference for the first derivative is:

$$\left.\frac{\partial u}{\partial x}\right|_{i,j} \approx \frac{-u_{i+2,j} + 8u_{i+1,j} - 8u_{i-1,j} + u_{i-2,j}}{12\,\Delta x} \tag{S.3}$$

.

In two dimensions, a common five-point stencil for the Laplacian $\nabla^2 u$ is:

$$\nabla^2 u_{i,j} \approx \frac{u_{i+1,j} - 2u_{i,j} + u_{i-1,j}}{\Delta x^2} + \frac{u_{i,j+1} - 2u_{i,j} + u_{i,j-1}}{\Delta y^2}. \tag{S.4}$$

Such stencils provide a flexible and systematic way to handle a variety of PDEs by reducing derivative calculations to simple arithmetic on grid values.

## B  THE ANATOMY OF FINO

In this section, we present the full statements of Proposition 1 and Theorem 1, as given in the main paper, together with their complete proofs. The proof of Proposition 1 relies on the following two lemmas.

**Lemma 3** (Composition Error Estimate). *Let $\Phi_{\Delta t}$ denote the exact evolution operator for time step $\Delta t$, such that*

$$u(t_{n+1}, \cdot) = \Phi_{\Delta t}\big(u(t_n, \cdot)\big), \quad where \ t_n = n\Delta t. \tag{S.5}$$

*Then, for every integer $n \geq 0$, we have*

$$u(t_n, \cdot) = \big(\Phi_{\Delta t}\big)^n \big(u(t_0, \cdot)\big), \tag{S.6}$$

*where $(\Phi_{\Delta t})^n$ denotes the $n$-fold composition of $\Phi_{\Delta t}$.*

*Proof.* We proceed by induction on $n$. Let $P(n)$ denote the desired statement.

**Base case ($n = 0$).** By definition, $t_0 = 0$, and thus

$$(\Phi_{\Delta t})^0\big(u(t_0, \cdot)\big) = u(0, \cdot) = u(t_0, \cdot). \tag{S.7}$$

Hence, $P(0)$ holds.

**Inductive step.** Assume the claim holds for some $k \geq 0$, i.e.,

$$u(t_k, \cdot) = (\Phi_{\Delta t})^k\big(u(t_0, \cdot)\big). \tag{S.8}$$

Then, for $k + 1$, we have

$$u(t_{k+1}, \cdot) = \Phi_{\Delta t}\big(u(t_k, \cdot)\big) = \Phi_{\Delta t}\Big((\Phi_{\Delta t})^k\big(u(t_0, \cdot)\big)\Big) = (\Phi_{\Delta t})^{k+1}\big(u(t_0, \cdot)\big). \tag{S.9}$$

Thus, $P(k + 1)$ also holds.

By the principle of mathematical induction, the statement is true for all $n \geq 0$. $\qquad\square$

**Lemma 4.** *Suppose $e_0 = 0$ and the sequence satisfies the recurrence*

$$e_{n+1} \leq C\,e_n + \varepsilon' \quad \text{for all } n \geq 0.$$

*Then, for every integer $k \geq 1$, we have*

$$e_k \leq \big(1 + C + C^2 + \cdots + C^{k-1}\big)\varepsilon' = \frac{C^k - 1}{C - 1}\,\varepsilon' \quad (\text{assuming } C \neq 1).$$

*Proof.* We proceed by induction on $k$.

**Base case ($k = 1$).** From the recurrence, we have

$$e_1 \leq C\,e_0 + \varepsilon' = C \cdot 0 + \varepsilon' = \varepsilon'.$$

This matches the claimed bound since $1 + C^{1-1} = 1$.

**Inductive step.** Assume the claim holds for some $k \geq 1$, i.e.,

$$e_k \leq \big(1 + C + C^2 + \cdots + C^{k-1}\big)\varepsilon'.$$

Then, using the recurrence,

$$e_{k+1} \leq Ce_k + \varepsilon' \leq C\big(1 + C + C^2 + \cdots + C^{k-1}\big)\varepsilon' + \varepsilon'.$$

Factorizing $\varepsilon'$ gives

$$e_{k+1} \leq \big(C + C^2 + \cdots + C^k + 1\big)\varepsilon' = (1 + C + C^2 + \cdots + C^k)\varepsilon'.$$

Thus the claim also holds for $k + 1$.

By induction, the bound holds for all integers $k \geq 1$. $\qquad\square$

**Proposition 5** (Local-to-Global Error Bound). *Let $\Psi_\theta$ and $\Phi_{\Delta t}$ be two maps on a Banach space satisfying:*

$$\big\|\Psi_\Theta(u) - \Phi_\Lambda(u)\big\| \leq \varepsilon' \quad \text{for all } u \text{ in the relevant norm and for some constants } \epsilon' \geq 0.$$

*(This comes from the fact that we can approximate each local map $\Phi_{\Delta t}$ by $\Psi_\Theta(u)$ to within $\epsilon'$) and Lipschitz, i.e.*

$$\big\|\Phi_{\Delta t}(v) - \Phi_{\Delta t}(w)\big\| \leq C\,\|v - w\| \quad \forall\, v, w,$$

*for some constants $\epsilon' \geq 0$ and $C > 0$. Define the iterates*

$$u_{k+1} = \Psi_\theta(u_k), \quad \tilde{u}_{k+1} = \Phi_{\Delta t}(\tilde{u}_k), \quad u_0 = \tilde{u}_0.$$

*Then after $K$ steps,*

$$\big\|(\Psi_\theta)^K(u_0) - (\Phi_{\Delta t})^K(u_0)\big\| \leq \frac{C^K - 1}{C - 1}\epsilon' \quad (\text{for } C \neq 1),$$

*, where $(\Psi_\theta)^K(u_0)$ means the $K$-fold composition of the map $\Psi_\Theta$ starting from $u_0$. and if $C = 1$ the right-hand side is simply $K\,\epsilon'$.*

*Proof.* By definition: $\|u_{k+1} - \tilde{u}_{k+1}\| = \|\Psi_\theta(u_k) - \Phi_{\Delta t}(\tilde{u}_k)\|$. We first rewrite

$$\Psi_\theta(u_k) - \Phi_{\Delta t}(\tilde{u}_k) = \big(\Psi_\theta(u_k) - \Phi_{\Delta t}(u_k)\big) + \big(\Phi_{\Delta t}(u_k) - \Phi_{\Delta t}(\tilde{u}_k)\big), \qquad \text{(S.10)}$$

By Triangle inequality and equation S.10 : We get

$$\|u_{k+1} - \tilde{u}_{k+1}\| \le \|\Psi_\theta(u_k) - \Phi_{\Delta t}(u_k)\| + \|\Phi_{\Delta t}(u_k) - \Phi_{\Delta t}(\tilde{u}_k)\|. \qquad \text{(S.11)}$$

By the assumption that $\|\Psi_\theta(u) - \Phi_{\Delta t}(u)\| \le \epsilon'$ and $\|\Phi_{\Delta t}(v) - \Phi_{\Delta t}(w)\| \le C\|v - w\|$. Equation S.11 can be further simplified as $\|u_{k+1} - \tilde{u}_{k+1}\| \le \epsilon' + C\|u_k - \tilde{u}_k\|$. Let $e_k = \|u_k - \tilde{u}_k\|$. Then

$$e_{k+1} \le C\,e_k + \epsilon'$$

. By Lemma 4, we get $e_k \le \big(1 + C + C^2 + \cdots + C^{k-1}\big)\varepsilon' = \frac{C^k - 1}{C - 1}\varepsilon'$   (assuming $C \ne 1$). After $K$ steps, we get

$$\big\|(\Psi_\Theta)^K(u_0) - (\Phi_\Lambda)^K(u_0)\big\| = e_K \le \frac{C^K - 1}{C - 1}\epsilon'.$$

$\square$

**Theorem 6** (Universal Approximation of FINO for Discrete Time–Stepped PDE Dynamics). *Let $G \subset \mathbb{R}^d$ be a finite spatial grid with $m$ nodes, and let*

$$X := \mathbb{R}^{m \times c}, \qquad \|u\| := \left(\frac{1}{mc}\sum_{i=1}^{m}\sum_{\ell=1}^{c}|u_{i,\ell}|^2\right)^{1/2},$$

*denote the Banach space of grid–based states. Let $\Phi_{\Delta t} : X \to X$ be the exact one–step evolution operator of a semi–discrete PDE, assumed to be Lipschitz stable:*

$$\|\Phi_{\Delta t}(v) - \Phi_{\Delta t}(w)\| \le C\|v - w\|, \qquad \forall v, w \in X, \qquad \text{(S.12)}$$

*for some constant $C \ge 1$.*

*For a final time $T = K\Delta t$ with integer $K \ge 1$, define the exact solution after $K$ steps as*

$$u(T) = \big(\Phi_{\Delta t}\big)^K(u_0),$$

*where $(\Phi_{\Delta t})^K$ denotes the $K$–fold composition.*

*Then for every compact set $\mathcal{U} \subset X$ and every tolerance $\varepsilon > 0$, there exists a depth–$K$ FD–NET*

$$\Psi_\theta^{(K)} := \underbrace{\Psi_\theta \circ \cdots \circ \Psi_\theta}_{K \text{ identical blocks}},$$

*such that*

$$\sup_{u_0 \in \mathcal{U}} \big\| \Psi_\theta^{(K)}(u_0) - \big(\Phi_{\Delta t}\big)^K(u_0)\big\| \le \varepsilon.$$

*In other words, the class of FINO is dense in the set of discrete solution operators of Lipschitz–stable PDEs on compact subsets of $X$.*

*Proof.* **Step 1 (Semigroup expansion).** By Lemma 3, the exact solution at time $t_n = n\Delta t$ is

$$u(t_n, \cdot) = \big(\Phi_{\Delta t}\big)^n\big(u(t_0, \cdot)\big). \qquad \text{(S.13)}$$

In particular, the target operator at time $T = K\Delta t$ is $\Phi_{\Delta t}^K$. Therefore, it suffices to approximate the one-step map $\Phi_{\Delta t}$ and then compose $K$ times.

**Step 2 (Compactness of the reachable set).** Define the compact "reachable" set

$$\mathcal{V} := \bigcup_{j=0}^{K-1}\big(\Phi_{\Delta t}\big)^j(\mathcal{U}).$$

Continuity of $\Phi_{\Delta t}$ implies by induction that each image $(\Phi_{\Delta t})^j(\mathcal{U})$ is compact, hence $\mathcal{V}$ is compact.

**Step 3 (One-step uniform approximation by an FINO block).** Since $X$ is finite-dimensional and $\Phi_{\Delta t} : \mathcal{V} \to X$ is continuous on the compact set $\mathcal{V}$, the classical Universal Approximation Theorem for feed-forward networks (e.g., Cybenko 1989; Hornik, Stinchcombe & White 1989) ensures that for any $\varepsilon' > 0$ there exists a (finite) neural network $F_\theta : \mathcal{V} \to X$ with

$$\sup_{u \in \mathcal{V}} \left\| F_\theta(u) - \Phi_{\Delta t}(u) \right\| \leq \varepsilon'. \tag{S.14}$$

An FD–NET one-step block $\Psi_\theta$ (a residual Euler update $u \mapsto u + \Delta t \, \mathcal{N}_\theta(u)$ with a finite convolutional/ReLU stack $\mathcal{N}_\theta$) is a special case of such a feed-forward map from $X$ to $X$. Hence, by increasing width/depth of the block, we can realize equation S.14 with $\Psi_\theta$ in place of $F_\theta$:

$$\sup_{u \in \mathcal{V}} \left\| \Psi_\theta(u) - \Phi_{\Delta t}(u) \right\| \leq \varepsilon'. \tag{S.15}$$

**Step 4 (Error recursion).** Let $u_0 = \tilde{u}_0 \in \mathcal{U}$ and define the two sequences

$$u_{k+1} = \Psi_\theta(u_k), \qquad \tilde{u}_{k+1} = \Phi_{\Delta t}(\tilde{u}_k), \qquad k = 0, 1, \ldots, K-1.$$

Set $e_k := \| u_k - \tilde{u}_k \|$. Adding and subtracting $\Phi_{\Delta t}(u_k)$ and using equation S.12 and equation S.15, we obtain

$$e_{k+1} = \left\| \Psi_\theta(u_k) - \Phi_{\Delta t}(\tilde{u}_k) \right\| \leq \underbrace{\left\| \Psi_\theta(u_k) - \Phi_{\Delta t}(u_k) \right\|}_{\leq \varepsilon'} + \underbrace{\left\| \Phi_{\Delta t}(u_k) - \Phi_{\Delta t}(\tilde{u}_k) \right\|}_{\leq C \, e_k} \leq C \, e_k + \varepsilon',$$

with $e_0 = 0$.

**Step 5 (Geometric accumulation).** By Lemma 4 (Geometric Error Lemma) and Proposition 5 , the recursion $e_{k+1} \leq C e_k + \varepsilon'$ with $e_0 = 0$ yields

$$e_k \leq \begin{cases} \dfrac{C^k - 1}{C - 1} \, \varepsilon', & C \neq 1, \\ k \, \varepsilon', & C = 1, \end{cases} \qquad \text{for all } k = 1, \ldots, K.$$

In particular,

$$\left\| \Psi_\theta^{(K)}(u_0) - \Phi_{\Delta t}^K(u_0) \right\| = e_K \leq \begin{cases} \dfrac{C^K - 1}{C - 1} \, \varepsilon', & C \neq 1, \\ K \, \varepsilon', & C = 1. \end{cases} \tag{S.16}$$

**Step 6 (Choice of the local tolerance).** Given any $\varepsilon > 0$, choose

$$\varepsilon' := \begin{cases} \varepsilon \, \dfrac{C - 1}{C^K - 1}, & C \neq 1, \\ \varepsilon / K, & C = 1. \end{cases}$$

By equation S.16, this guarantees $e_K \leq \varepsilon$ uniformly for all $u_0 \in \mathcal{U}$. Since equation S.15 can be enforced by increasing the size of the one-step FINO block, the theorem follows. □

## C  Experiment Results

### C.1  Dataset and Implementation Detail

⭐ **1D Advection**
**Governing PDE.**

$$\partial_t u(t, x) + \beta \, \partial_x u(t, x) = 0, \qquad (t, x) \in (0, 2] \times (0, 1). \tag{S.17}$$

**Initial data.**

$$u(0, x) = u_0(x), \qquad x \in (0, 1). \tag{S.18}$$

**Parameters.** $\beta \in \mathbb{R}$ is the constant transport (advection) speed and we choose $\beta = 4$.

**Closed-form solution.** By the method of characteristics, the solution is a rigid shift of the initial profile:

$$u(t, x) = u_0(x - \beta t).$$ (S.19)

**Discretization and split.**

- Spatial grid: Offical $1024 \times 1024$ and a downsample factor by 8
- Temporal samples: 200 snapshots; first 10 used as inputs, remaining 190 as prediction targets and downsample a factor by 5.
- Dataset split: 9000 training / 1000 testing samples.

★ **1D Diffusion–Reaction**
**Governing PDE.**

$$\partial_t u(t, x) = \nu \, \partial_{xx} u(t, x) + \rho \, u(t, x)(1 - u(t, x)), \qquad x \in (0, 1), \; t \in (0, 1],$$ (S.20)

with initial data

$$u(0, x) = u_0(x), \qquad x \in (0, 1),$$ (S.21)

and periodic boundary conditions on $[0, 1]$:

$$u(t, 0) = u(t, 1), \qquad \partial_x u(t, 0) = \partial_x u(t, 1), \qquad t \in (0, 1].$$ (S.22)

**Dynamics.** The reaction term $\rho \, u(1 - u)$ can drive near–exponential transients, producing fast time–scale phenomena that stress both numerical solvers and learning surrogates.

**Initialization.** To avoid ill-posed or degenerate starts, the prescribed profile is rectified and normalized:

$$u_0(x) \; \leftarrow \; \frac{|u_0(x)|}{\max_{x \in (0,1)} |u_0(x)|},$$

so that $u_0 \in [0, 1]$ and $\|u_0\|_\infty = 1$.

**Discretization and split.**

- Spatial grid: Offical $1024 \times 1024$ and a downsample factor by 8
- Temporal samples: 200 snapshots; first 10 used as inputs, remaining 190 as prediction targets and downsample a factor by 5.
- Dataset split: 9000 training / 1000 testing samples.

★ **1D CNS**
The equations governing compressible fluid dynamics describe the evolution of density, momentum, and energy of a fluid system. They are written as

$$\partial_t \rho + \nabla \cdot (\rho \mathbf{v}) = 0,$$ (S.23)

$$\rho \big( \partial_t \mathbf{v} + \mathbf{v} \cdot \nabla \mathbf{v} \big) = -\nabla p + \eta \Delta \mathbf{v} + \big( \zeta + \tfrac{\eta}{3} \big) \nabla (\nabla \cdot \mathbf{v}),$$ (S.24)

$$\partial_t \Big( \epsilon + \tfrac{\rho \|\mathbf{v}\|^2}{2} \Big) + \nabla \cdot \Big[ \big( \epsilon + p + \tfrac{\rho \|\mathbf{v}\|^2}{2} \big) \mathbf{v} - \mathbf{v} \cdot \boldsymbol{\sigma}' \Big] = 0,$$ (S.25)

For the details of the notation and description, we refer to PDEBench (Takamoto et al., 2022)

**Discretization and split.**

- Spatial grid: Offical $1024 \times 1024$ and a downsample factor by 8
- Temporal samples: 100 snapshots; first 10 used as inputs, remaining 90 as prediction targets and downsample a factor by 5.
- Dataset split: 9000 training / 1000 testing samples.

### ★ 2D Darcy Flow

**Governing PDE.** On the unit square $\Omega = (0,1)^2$, the steady 2D Darcy flow is

$$-\nabla\cdot\big(a(x,y)\,\nabla u(x,y)\big) = f(x,y), \quad (x,y) \in \Omega, \tag{S.26}$$

with homogeneous Dirichlet boundary condition

$$u(x,y) = 0, \qquad (x,y) \in \partial\Omega. \tag{S.27}$$

Here $a(x,y)$ denotes the diffusion coefficient and $u(x,y)$ the solution field.

**Operator-learning objective.** We aim to learn the solution operator

$$\mathcal{S}: a \mapsto u, \qquad (x,y) \in \Omega, \tag{S.28}$$

so that, given $a$, the predictor returns the corresponding solution $u$ of equation S.26–equation S.27.

**Data protocol.** Following the PDEBench setup Takamoto et al. (2022):

- *Forcing.* A spatially uniform load $f(x,y) \equiv \beta$ with $\beta = 1.0$.
- *Splits.* 9000 training samples and 1000 test samples.
- *Resolution.* Fields are provided on the official grid $128 \times 128$ and downsampled by a factor of two.

### ★ 2D Shallow–Water

**Governing PDEs.**

$$\partial_t h + \partial_x(hu) + \partial_y(hv) = 0, \tag{S.29}$$

$$\partial_t(hu) + \partial_x\big(u^2 h + \tfrac{1}{2}g_r h^2\big) = -g_r\, h\, \partial_x b, \tag{S.30}$$

$$\partial_t(hv) + \partial_y\big(v^2 h + \tfrac{1}{2}g_r h^2\big) = -g_r\, h\, \partial_y b. \tag{S.31}$$

**State and coefficients.**

- $h(x,y,t)$: water depth.
- $(u(x,y,t),\, v(x,y,t))$: depth-averaged velocities in the $x$- and $y$-directions.
- $b(x,y)$: bathymetry (spatially varying bed elevation).
- $g_r$: gravitational acceleration.

**Domain and time horizon.**

- Spatial domain $\Omega = [-2.5,\, 2.5]^2$.
- Time interval $t \in [0,1]\,$s.

**Initial condition (radial dam–break).**

$$h(0,x,y) = \begin{cases} 2.0, & \sqrt{x^2+y^2} < r, \\ 1.0, & \sqrt{x^2+y^2} \geq r, \end{cases} \qquad r \sim \mathcal{U}(0.3,\, 0.7).$$

**Learning objective (solution operator).**

$$\mathcal{S}: h\big|_{t\in[0,t']} \longmapsto h\big|_{t\in[t',T]}, \qquad (x,y) \in \Omega,$$

with $t' = 0.009\,$s and $T = 1.000\,$s.

**Discretization and split.**

- Spatial grid: Offical $128 \times 128$ and downsample factor by 2.
- Temporal samples: 101 snapshots; first 10 used as inputs, remaining 91 as prediction targets.
- Dataset split: 900 training / 100 testing samples.

Table S.1: Comparison of baselines and FINO on 1D PDE benchmarks—advection, compressible Navier–Stokes (CNS), and diffusion–reaction — measured by normalised RMSE (nRMSE) , max error and conservation RMSE (cRMSE). The best results are highlighted in green .

| METHOD | advection (1D) | | | CNS (1D) | | | diffusion-reaction (1D) | | |
|---|---|---|---|---|---|---|---|---|---|
| | nRMSE↓ | max error↓ | cRMSE↓ | nRMSE↓ | max error↓ | cRMSE↓ | nRMSE↓ | max error↓ | cRMSE↓ |
| UNet | 0.09246 | 0.50268 | 0.03269 | 0.8123 | 50.83448 | 15.34809 | 0.07399 | 0.21527 | 0.05359 |
| FNO | **0.00892** | **0.11371** | **0.00032** | 0.23532 | 4.78869 | 0.0879 | 0.05484 | 0.0521 | 0.03036 |
| UFNO | 0.0175 | 0.26239 | 0.00129 | 0.37418 | 5.42717 | 0.17281 | 0.01705 | 0.02608 | 0.01026 |
| FFNO | 0.01198 | 0.17062 | 0.00083 | **0.16448** | **4.55828** | **0.06288** | 0.02685 | **0.0304** | 0.01511 |
| Trasnsovler | 0.01555 | 0.17913 | 0.00083 | 0.35738 | 8.22922 | 0.22427 | **0.01347** | 0.03871 | **0.00963** |
| FINO | 0.0049 | 0.09395 | 0.00036 | 0.14635 | 3.53841 | 0.0654 | 0.01137 | 0.04157 | 0.00729 |
| Improvement | 45.07 % | 17.38 % | 12.5 % | 11.02% | 22.37% | 4.01% | 15.60 | -59.4 | 24.30 |

★ **2D Diffusion–Reaction**

**State variables.** $u = u(x, y, t)$ (activator), $v = v(x, y, t)$ (inhibitor).

**Governing PDEs.**

$$\partial_t u = D_u \left( \partial_{xx} + \partial_{yy} \right) u + R_u(u, v), \tag{S.32a}$$
$$\partial_t v = D_v \left( \partial_{xx} + \partial_{yy} \right) v + R_v(u, v). \tag{S.32b}$$

**Reaction terms.**

$$R_u(u, v) = u - u^3 - k - v, \tag{S.33a}$$
$$R_v(u, v) = u - v. \tag{S.33b}$$

**Parameters and domain.**

- Diffusion coefficients: $D_u = 1 \times 10^{-3}, \quad D_v = 5 \times 10^{-3}$.

- Coupling constant: $k = 5 \times 10^{-3}$.

- Space–time: $\Omega = [-1, 1]^2, \quad t \in [0, 5]$.

**Operator learning target.**

$$\mathcal{S} : \{u, v\}_{t \in [0, t']} \longmapsto \{u, v\}_{t \in (t', T]}, \quad (x, y) \in \Omega, \tag{S.34}$$

with $t' = 0.045 \, \text{s}$ and $T = 5.000 \, \text{s}$.

**Discretization and splits.**

- Spatial grid: Official grid $128 \times 128$ and a downsample factor by 2.

- Temporal resolution: 101 frames (inputs: 10; prediction horizon: 91).

- Dataset size: 900 training and 100 testing trajectories, following the PDEBench protocol (Takamoto et al., 2022).

★ **2D Climate Modelling** (Kissas et al., 2022) use daily global fields from the NOAA PSL NCEP/NCAR Reanalysis to construct a paired dataset of near-surface air temperature (input) and surface pressure (target). Samples span ten calendar years split into two five-year blocks (2000–2004 train; 2005–2009 test), with leap days removed, on a co-registered $72 \times 72$ latitude–longitude grid covering $[-90°, 90°] \times [0°, 360°]$. The task is to learn a black-box operator

$$\mathcal{G} : C(X, \mathbb{R}) \to C(X, \mathbb{R})$$

mapping temperature to pressure for each day. We provide recommended preprocessing (regridding, normalization, area weighting), evaluation metrics, and caveats regarding physical ill-posedness and topographic effects.

Table S.2: Comparison of baselines and FINO on 2D PDE benchmarks—advection, compressible Navier–Stokes (CNS), and diffusion–reaction — measured by normalised RMSE (nRMSE) , max error and conservation RMSE (cRMSE). The best results are highlighted in green .

| METHOD | Darcy Flow (2D) | | | Diffusion Reaction (2D) | | | Shallow Water (2D) | | |
|---|---|---|---|---|---|---|---|---|---|
| | nRMSE↓ | max error↓ | cRMSE↓ | nRMSE↓ | max error↓ | cRMSE↓ | nRMSE↓ | max error↓ | cRMSE↓ |
| UNet | 0.05412 | 0.1699 | 0.01352 | 0.87007 | 0.2265 | 0.02115 | 0.11552 | 0.75481 | 0.03397 |
| FNO | 0.13772 | 0.22714 | 0.02368 | 0.19243 | 0.09539 | 0.00154 | 0.01393 | 0.13625 | 0.00058 |
| UFNO | 0.39773 | 0.56864 | 0.05821 | 0.23746 | 0.06644 | 0.00617 | 0.01518 | 0.2161 | 0.00074 |
| FFNO | 0.13121 | 0.25059 | 0.01976 | 0.19817 | 0.14103 | 0.00829 | 0.01145 | 0.12899 | 0.00069 |
| Trasnsovler | 0.10777 | 0.26815 | 0.0144 | 0.24379 | 0.10434 | 0.00869 | 0.00565 | 0.12001 | 0.00066 |
| LocalFNO | 0.10662 | 0.21291 | 0.02042 | **0.05492** | **0.05488** | **0.00077** | 0.00422 | 0.09241 | 0.00038 |
| FINO | **0.05764** | **0.16731** | **0.01252** | 0.03645 | 0.03772 | 0.00014 | 0.0025 | 0.03976 | 0.00013 |
| Improvement | -6.50% | -1.52% | -7.4% | 33.63% | 31.27% | 81.82% | 40.76 | 56.97 | 65.79 |

Table S.3: Comparison of baselines and FINO on PDEBench— compressible Navier–Stokes (CNS), 2D Diffusion–Reaction and Shallow Water — measured by RMSE in Fourier space, low frequency regime (fRMSEL) , RMSE in Fourier space, middle frequency regime (fRMSEM) and RMSE in Fourier space, high frequency regime (fRMSEH); lower is better. The best results are highlighted in green while the second best results are in bold font.

| METHOD | CNS (1D) | | | Diffusion Reaction (2D) | | | Shallow Water (2D) | | |
|---|---|---|---|---|---|---|---|---|---|
| | fRMSEL↓ | fRMSEM↓ | fRMSEH↓ | fRMSEL↓ | fRMSEM↓ | fRMSEH↓ | fRMSEL↓ | fRMSEM↓ | fRMSEH↓ |
| UNet | 4.42510 | 0.18985 | 0.03155 | 0.01446 | 0.00605 | 0.00153 | 0.03164 | 0.00874 | 0.00197 |
| FNO | 0.11258 | 0.04998 | 0.00706 | 0.00161 | 0.00120 | 0.00064 | 0.00060 | 0.00067 | 0.00148 |
| UFNO | 0.15604 | 0.05499 | 0.00682 | 0.00393 | 0.00127 | 0.00036 | 0.00300 | 0.00153 | 0.00078 |
| FFNO | 0.09931 | 0.04275 | **0.00686** | 0.00361 | 0.00115 | 0.00044 | 0.00224 | 0.00109 | 0.00043 |
| Trasnsovler | 0.27045 | 0.08406 | 0.01283 | 0.00393 | 0.0015 | 0.00053 | 0.00074 | 0.00061 | 0.00043 |
| LocalFNO | - | - | - | **0.00053** | **0.00044** | **0.00021** | 0.00042 | 0.00050 | 0.00033 |
| FINO | 0.06596 | 0.03021 | 0.00864 | 0.00024 | 0.00026 | 0.00014 | 0.00013 | 0.00027 | 0.00022 |
| Improvement | 33.58% | 29.33% | -26.69% | 54.72% | 40.91% | 33.33% | 69.05% | 46% | 33.33% |

## C.2 ADDITIONAL NUMERICAL RESULTS AND VISULISATIONS

Table S.1 compares baseline methods (U-Net, FNO, UFNO, FFNO, Transolver) with the proposed FINO model on three 1D PDE benchmarks: advection, compressible Navier–Stokes (CNS), and diffusion–reaction. Evaluation is based on three complementary error metrics: normalized RMSE (nRMSE), which ensures scale independence; maximum error, which captures the local worst-case discrepancy and serves as a proxy for stability in time-stepping; and conservation RMSE (cRMSE), which measures the deviation from conserved physical quantities. Across all tasks, FINO consistently outperforms baselines by large margins. For advection, FINO achieves the lowest nRMSE (0.0049), reducing error by 45.07% relative to the strongest baseline, while also improving stability and conservation. In CNS, FINO delivers the best nRMSE (0.146) and cRMSE (0.065), yielding 11.02% and 22.37% improvements, respectively. In the diffusion–reaction system, FINO again provides the lowest nRMSE (0.01137) and cRMSE (0.00729), with a 24.30% gain. These results demonstrate that FINO not only improves accuracy but also enhances stability and preserves key physical invariants across disparate PDE regimes.

Table S.2 reports results on three challenging 2D PDE benchmarks—Darcy Flow, diffusion–reaction, and Shallow Water—using normalized RMSE (nRMSE) for scale-independent accuracy, maximum error as a proxy for stability, and conservation RMSE (cRMSE) to quantify deviations from conserved physical quantities. FINO consistently achieves state-of-the-art performance across all tasks. For Darcy Flow, it attains second-best results closely behind UNet. Since Darcy flow is a time-independent PDE and other global method or Local plus Global method perform poorly. In contrast, FINO achieves a very similar performance with UNet. In 2D diffusion–reaction, FINO delivers substantial gains with the lowest errors across all three metrics (nRMSE = 0.03645, cRMSE = 0.00014), surpassing LocalFNO by up to 81.8% in conservation accuracy. In 2D shallow water, FINO again outperforms all baselines by a wide margin, achieving nRMSE = 0.0025, maximum error = 0.0398 and cRMSE = 0.00013, reflecting improvements of 40–65%. Since 2D Diffusion Reaction and Shallow Water both have a long time domain, such strong performance indicates FINO can achieve a stable and accurate result in a long time domain. These results highlight FINO's robustness across diverse PDE regimes, combining local accuracy, numerical stability, and strong adherence to physical conservation laws.

## 2D Climate Modelling

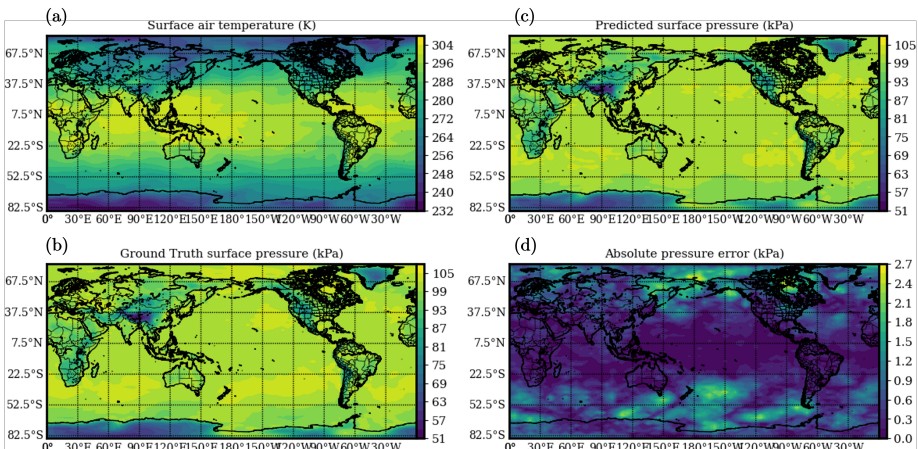

Figure S.1: Comparison of full-resolution predictions and baseline on the climate modeling benchmark. (a) Input surface air temperature field. (b) Ground-truth surface pressure. (c) Predicted surface pressure from FINO. (d) Absolute error map between prediction and ground truth.

Table S.3 evaluates baselines and FINO on PDEBench using Fourier space errors across three regimes—low (fRMSEL), middle (fRMSEM), and high frequency (fRMSEH)—to measure the fidelity of capturing multi-scale dynamics. In contrast with RMSE and nRMSE, which provide a metric view of the data. These Fourier space errors provide a physical view. Lower values indicate better accuracy in representing frequency components. On CNS, FINO delivers the best performance in the low and middle frequency bands, reducing error by 33.6% and 29.3% compared to the strongest baselines, though it is slightly less competitive in the high-frequency regime. For 2D diffusion–reaction, FINO achieves substantial improvements across all bands, with an 81.8% relative gain in the high-frequency regime, highlighting its ability to preserve fine-scale oscillatory structures. In the Shallow Water system, FINO again attains the lowest errors across all frequency bands, with improvements of 69% in low frequencies, 46% in middle frequencies, and 33% in high frequencies, demonstrating superior resolution of both large-scale flows and small-scale turbulent components. These results emphasize FINO's strong capacity to resolve multi-scale PDE dynamics in Fourier space, outperforming both spectral and local baselines across diverse regimes.

## D    USE OF LARGE LANGUAGE MODELS (LLMS)

During the preparation of the paper, LLMs were used to polish part of the writing.

# 1D advection

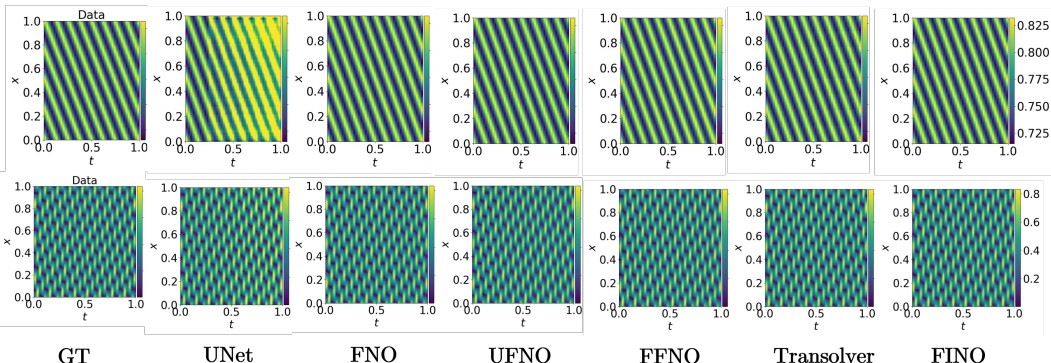

Figure S.2: Visualization of 1D advection across different baselines on two samples.

# 2D Diffusion-Reaction (activator)

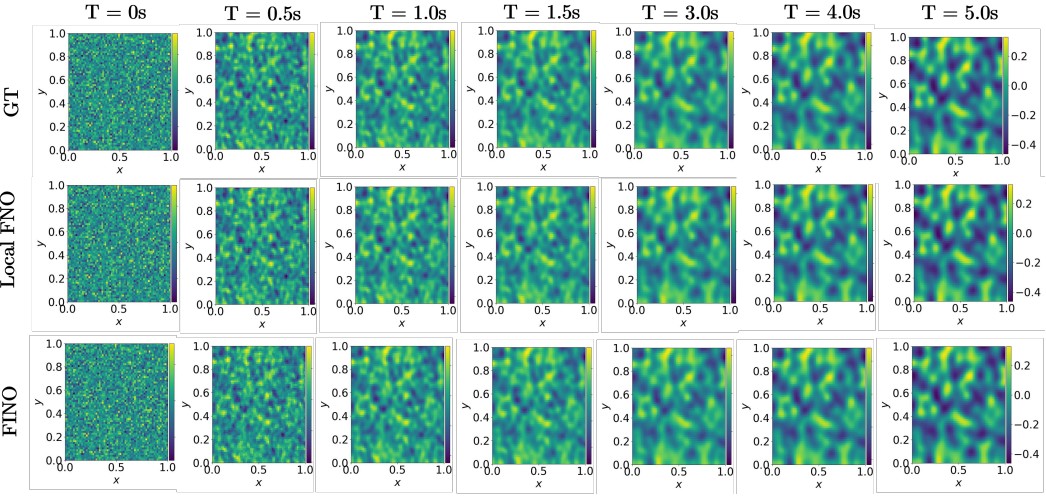

Figure S.3: Qualitative Comparison of 2D Diffusion–Reaction (Activator) Across GT, Local FNO, and our method

# 2D Diffusion-Reaction (inhibitor)

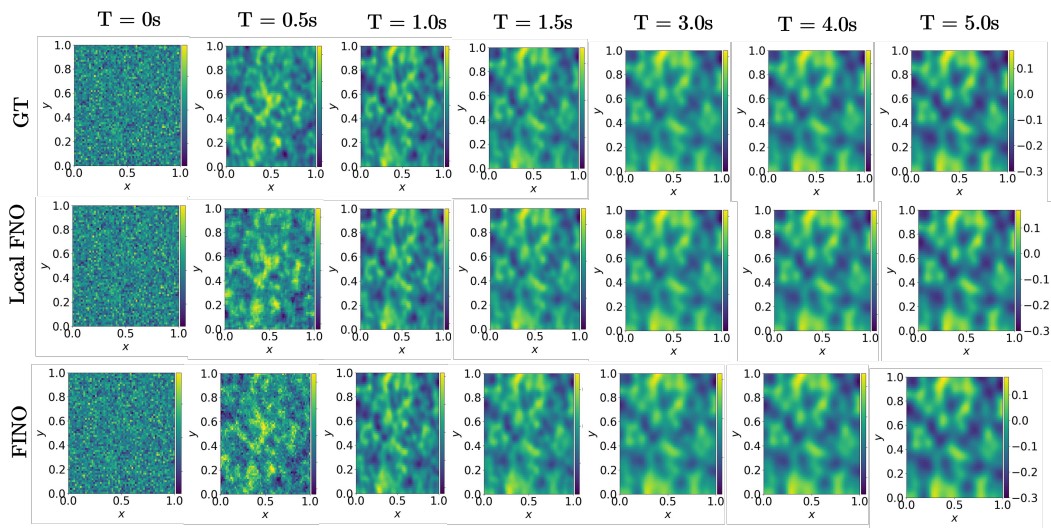

Figure S.4: Qualitative Comparison of 2D Diffusion–Reaction (inhibitor) Across GT, Local FNO, and our method

