# OpenReview forum: "PDE Solvers Should Be Local: Fast, Stable Rollouts with Learned Local Stencils"
_ICLR.cc/2026/Conference — ICLR 2026 Conference Withdrawn Submission_

### Official Review · Reviewer_afhf · 2025-10-28

**Soundness:** 2
**Presentation:** 3
**Contribution:** 1
**Rating:** 2
**Confidence:** 3

**Summary:**

This paper proposes a neural PDE surrogate model. The main building block of the model is composed of convolution, gating and time integration. Additionally, the model uses a downsample - upsample structure as in UNet. The model is validated on PDEBench and weather forecasting.

**Strengths:**

- The proposed model mostly achieves better performance compared to other reported baselines.

**Weaknesses:**

- There is no ablation studies on the main design choices, namely the stencil operator, the gating, and time-integration using small / learnable time intervals.
- The model uses a UNet architecture, which learns global features. The effect of using local operators is not very clearly demonstrated.

**Questions:**

- In Equation 1, what is the difference between $S(\mathbf{u})$ and a conventional convolution?
- Is section 3.2 specific to FINO or is applicable to any neural operators?

---

### Official Review · Reviewer_iZfd · 2025-10-30

**Soundness:** 2
**Presentation:** 2
**Contribution:** 1
**Rating:** 2
**Confidence:** 4

**Summary:**

The paper proposes a new architecture, FINO (Finite-difference inspired neural operator), for solving partial differential equations (PDEs). The authors argue that existing state-of-the-art models, such as FNOs or Transformers, rely on global mixing mechanisms (via spectral convolutions or attention), which can be a poor inductive bias for many physical systems (oversmoothing sharp, local dynamics or incurring unnecessarily high computational costs). Inspired by classical finite-difference methods and U-Nets, FINO consists of learnable convolutional kernels to approximate spatial derivatives, which are used in an explicit, learnable time-stepping scheme (mimicking a forward Euler method) to evolve the system state. FINO is evaluated on six PDEBench benchmarks and a climate modeling task, where it can outperform baselines in terms of speed and accuracy.

**Strengths:**

The paper provides arguments why locality can be an important inductive bias for certain classes of PDEs. Then the authors draw inspiration from local numerical schemes, i.e., finite-difference schemes combined with a forward Euler step, for designing FINO to explicitly learn spatial derivatives and a temporal integration step. The performance/cost-tradeoff in the numerical experiments and the ablation study on data scaling, suggest that this can act as a good inductive bias.

**Weaknesses:**

- The central building block of the architecture is described as a "differential stencil layer, a gating mask, and a linear fuse step", which in practice results in a rather standard gated-convolutional block. Moreover, there is a plethora of related works that recognize the connection of convolutional filters and finite-difference methods and propose variants of learnable finite-difference methods, e.g., https://arxiv.org/abs/2201.01854, https://arxiv.org/abs/2002.03014, https://arxiv.org/abs/2006.01892, https://arxiv.org/abs/2311.00259. While https://arxiv.org/pdf/1710.09668 is mentioned in the related works, it is unclear why it is not used as another baseline.
- FINO is not a neural *operator*, since it is not discretization invariant (the convolutional kernels converge to a pointwise operator in the limit of increasing resolutions). In particular, the provided approximation result in Theorem 6 also only holds for a fixed resolution.
- The time-step $\Delta t$ is learnable, although the time-step seems to be prescribed by the data. Moreover, explicit time-stepping schemes, like the forward Euler method, are only stable under certain conditions (e.g., the CFL condition) and the maximum stable $\Delta t$ is linked to the spatial grid size and the structure of the PDE (in particular, different initial conditions or PDE parameters).
- The "novel error-propagation bound" is a standard tool for bounding the composition error of numerical methods (discrete version of a Grönwall's inequality). In general, the theoretical results follow directly from the universal approximation theorem and a bound on geometric series and do not seem to be specific to the proposed architecture or the claim that PDE solvers should be local (which generally seems to be too strong of a claim). Moreover, there is no guarantee (and also no empirical evaluation) that the Lipschitz constant $C$ (in particular dependent on the learnable $\Delta t$) is sufficiently small.
- Details are missing on how the baselines and the proposed method have been tuned. Some of the baseline numbers seem to be lower than reported in the respective papers, e.g., Diffusion Reaction for LocalFNO. On the PDEBench datasets, there are also significantly stronger baselines, see, e.g., https://arxiv.org/pdf/2403.03542 Table 1 for a comparison.

**Minor:**
- It would be interesting to have a reference for the statement “Global operator methods often perform poorly on time-independent PDEs, limiting their applicability”
- Why is FFNO not evaluated on the Climate Modelling dataset?
- Since the visualization results look very similar, it might be better to visualize the error to the ground truth.
- Typo: “Climte Modelling”

**Questions:**

Concerns are mentioned in "weaknesses" above.

---

### Official Review · Reviewer_eU8k · 2025-10-30

**Soundness:** 1
**Presentation:** 2
**Contribution:** 1
**Rating:** 2
**Confidence:** 4

**Summary:**

This paper introduces FINO (Finite-difference-inspired Neural Operator), a neural operator architecture that enforces strict locality through learnable convolutional stencils and explicit time-stepping. The authors argue that global mixing mechanisms in existing neural operators (like FNO and Transformers) tend to oversmooth sharp local dynamics and incur high computational costs. FINO replaces fixed finite-difference coefficients with learnable convolutional kernels and evolves states via an explicit, learnable Euler scheme. Extensive experiments across six PDEBench problems and a climate modeling task demonstrate FINO achieves up to 44% lower error and 2x speedups over state-of-the-art baselines.

**Strengths:**

The model exhibits excellent performance on the dataset in comparison with the established baselines。

**Weaknesses:**

1.The use of finite difference convolutional kernels as a substitute for fixed stencils is not a novel concept, as similar research has already been conducted in models like PDENet and PeRCNN. Notably, these foundational models are not mentioned in the current work.

2.The strict convolutional design inherently assumes regular grids, limiting applicability to problems with complex geometries or irregular domains where graph-based or mesh-adaptive methods excel.

3.The forward Euler scheme, while interpretable, may face stability constraints (CFL conditions) for stiff problems, though this isn't thoroughly analyzed.

4.Some recent local operator methods (e.g., CNO, PDE-Net, PeRCNN) receive limited discussion in the related work.

**Questions:**

1.The explicit Euler scheme is known to have stability limitations for stiff PDEs. Did you encounter any stability issues during training or long rollouts, and how does the learnable time step parameter interact with traditional CFL conditions?

2.The learned convolutional kernels replace traditional finite-difference stencils. Can you provide insights into what these kernels learn - do they resemble classical stencils, or discover novel discretizations?

3.Given the strict convolutional design, how might FINO be extended to handle irregular geometries or adaptive meshes? Would you consider hybrid approaches with graph-based methods for complex domains?

4.The U-Net encoder-decoder provides multi-scale processing, but how does this interact with the strictly local stencil operations? Are there concerns about information loss during downsampling for problems requiring high-frequency preservation?

5.How sensitive is FINO's performance to key hyperparameters like stencil size, number of blocks, and the gating mechanism parameters? Are there guidelines for setting these for new PDE families?

6.How does FINO compare against traditional finite-difference solvers (with optimized stencils) in terms of both accuracy and computational efficiency, particularly for problems where high-order schemes are beneficial?

---

### Official Review · Reviewer_YAgu · 2025-10-31

**Soundness:** 3
**Presentation:** 3
**Contribution:** 2
**Rating:** 4
**Confidence:** 4

**Summary:**

The authors introduce an architecture for learning time-dependent PDEs. They design their architecture such that they learn filters in the same way finite-difference schemes operate in classical PDE solvers. Their architecture uses a gating mechanism to select which "derivatives" to apply to the input as well as a  learned timestep for advancing the solution in time. Moreover, their representation is fed through a bottleneck and then downsampled and upsampled like in a U-Net to capture changes at all scales. The authors conduct a  theoretical analysis where they they show that under some conditions on the Lipschitz constant of the ground-truth function, that the error compounds sub-linearly with time. Their experiments on PDE Bench and Climate modeling show strong results compared to the baseline and that their approach is fast when it comes to training their model as well as low inference time.

**Strengths:**

- Results show strong performances compared to the baselines in addition to being lightweight for training and inference.
- Experiments conducted on a wide variety of PDEs in addition to experiments on climate modeling.
- The paper presents theoretical analysis of the error propagation through time.

**Weaknesses:**

The paper introduces a Neural Operator architecture yet there are no experiments to show the architecture's performance as the resolution is increased as is customary in neural operator papers. A frequency analysis of the predictions would also be desirable to show what happens to the high-frequency content in the predictions and how much they differ from the ground-truth.
- Beyond the downstream results, it's hard to see how exactly the architecture gets better performance. It looks like a CNN with bells and whistles so it would be good to ablate the architecture to pinpoint exactly what makes it work. For example, $\Delta t$ seems to act as a learnable skip-connection, what happens if you fix its value to $\Delta t=1$ like a normal skip connection? What happens if there's no gating?
- Since the convolutions are supposed to mimic finite-difference schemes, it would be good to have some analysis of the learned filters (by visualizing them for example) and relating them to known finite-difference filters.
- Theorem 1 shows that if $C>1$ the bound is not very informative since it will increase exponentially. Do the example PDEs considered in the paper all have maps that have a Lipschitz constant of $C\leq 1$ ? If not what happens to the error?  What about your architecture's Lipschitz constant, are there any guarantees that its constant would be $\leq 1$ ? Because otherwise, the initial assumption that the maps would be below some $\epsilon$ wouldn't hold. The title claims that "PDE SOLVERS SHOULD BE LOCAL: FAST, STABLE ROLLOUTS WITH LEARNED LOCAL STENCIL" yet there are no experiments showing how the error evolves in time and those experiments would be a good opportunity to see if the error is indeed bounded by the bound you provided.

**Questions:**

- The uncertainty principle is mentioned a lot in the paper, yet to the reader unfamiliar with such a principle it's unclear. It would be good to add a subsection explaining what it is and what it implies.
- Typo in line 322: Climte -> Climate.
- How is $\Delta t$ constrained to be positive?
- In equation (9), $\mathcal{U}$ is used but not defined, same as $\mathcal{D}$.
- Figure 1 needs to be re-done as it's not very clear how the operations described in the paper related to the blocks in the architecture diagram. Be as explicit as possible.
- It would be good to see the effect of the convolution filter size.
- Figure 5 should also contain the error.

---

### Note · Authors · 2025-11-15

I have read and agree with the venue's withdrawal policy on behalf of myself and my co-authors.